# Effect of a Diet Supplemented with Nettle (*Urtica dioica* L.) or Fenugreek (*Trigonella foenum-graecum* L.) on the Post-Slaughter Traits and Meat Quality Parameters of Termond White Rabbits

**DOI:** 10.3390/ani11061566

**Published:** 2021-05-27

**Authors:** Sylwia Ewa Pałka, Agnieszka Otwinowska-Mindur, Łukasz Migdał, Michał Kmiecik, Dorota Wojtysiak

**Affiliations:** Department of Genetics, Animal Breeding and Ethology, University of Agriculture in Krakow, al. Mickiewicza 24/28, 30-059 Krakow, Poland; agnieszka.otwinowska@urk.edu.pl (A.O.-M.); lukasz.migdal@urk.edu.pl (Ł.M.); michal.kmiecik@urk.edu.pl (M.K.); dorota.wojtysiak@urk.edu.pl (D.W.)

**Keywords:** rabbit, post-slaughter traits, meat quality traits, nettle (*Urtica dioica* L.), fenugreek (*Trigonella foenum-graecum* L.)

## Abstract

**Simple Summary:**

Herbs can be a perfect supplement in the animal diet. In addition, they improve the functioning of the immune system, regulate the appetite and feed intake of animals, regulate the functioning of the digestive system, stimulate the metabolism of the body as well as quality parameters of meat. In the literature regarding herbal supplements to rabbit fodder, there is not much information on the use of nettle or fenugreek in their nutrition. Both of these herbs are valuable sources of vitamins and minerals. They regulate the digestive system and stimulate appetite. They have a positive impact on the functioning of the immune system and they exhibit antibacterial activity. Nettle improves biochemical, haematological, and immunological parameters. Numerous studies have shown that fenugreek can also have anti-inflammatory, analgesic, and antipyretic properties. The purpose of the study was to analyse the effect of nettle (*Urtica dioica* L.) leaves and fenugreek (*Trigonella foenum-graecum* L.) seeds as additives to fodder in order to improve post-slaughter traits and quality parameters Termond White of rabbit meat. Rabbits fed with pellets with the addition of nettle were characterized by a higher slaughter weight, higher weight of hot and cold carcasses, as well as a higher weight of the fore, middle and hind part of the carcass compared to the other two groups. The carcasses of animals fed with fenugreek and nettle had a higher percentage of the fore and hind parts compared to the carcasses of the animals from the control group. For most colour measurement traits, the differences depending on the feeding regime were significant. Summing up, it can be stated that feeding rabbits with herbal supplements, such as nettle or fenugreek, has no negative impact on technological parameters of the meat or the microstructure of the muscles. On the contrary, in the case of nettle, one can speak of a beneficial effect of this herbal supplement on the post-slaughter traits and microstructure of the muscles, as evidenced by the greater share of type I muscle fibres in the composition of muscles, e.g., *longissimus lumborum*, indicating a more oxidative nature of these muscles, which in turn may translate into the quality of rabbit meat.

**Abstract:**

The purpose of the study was to analyse the effect of nettle (*Urtica dioica* L.) leaves and fenugreek (*Trigonella foenum-graecum* L.) seeds as additives to fodder in order to improve post-slaughter traits and quality parameters of Termond White rabbit meat (n = 60; 30♂, 30♀). Three experimental groups were created. The control group (n = 20; 10♂ and 10♀) was fed ad libitum feed containing corn, bran, wheat, dried alfalfa, soybean meal, sunflower meal, dicalcium phosphate, calcium carbonate and vitamin-mineral premix. The animals from the first experimental group (n = 20; 10♂ and 10♀) were fed a complete mixture added with 1% of nettle (*Urtica dioica* L.) leaves. Rabbits from the second group (n = 20; 10♂ and 10♀) were fed with a complete mixture added with 1% of fenugreek (*Trigonella foenum-graecum* L.) seeds. Rabbits fed with pellets with the addition of nettle were characterized by a higher slaughter weight, higher weight of hot and cold carcasses, lungs, kidneys and head as well as a higher weight of the fore, middle and hind part of the carcass compared to the other two groups. The carcasses of animals fed with fenugreek and nettle had a higher percentage of the fore and hind parts compared to the carcasses of the animals from the control group. The female carcasses were characterized by a significantly higher percentage of the middle part compared to the male carcasses. For most colour measurement traits, the differences depending on the feeding regime were significant. The effect of gender on meat colour was non significant. The effect of feeding regime and of gender on texture traits such as shear force, hardness, springiness, cohesiveness and chewiness were non significant. Feeding had no effect on muscle fibre diameter, but it affected the muscle fibre type I percentage. Thus, the group fed with pellets containing nettle leaves had higher percentage of type I muscle fibres than the control group. The effect of gender on muscle fibre traits was non significant.

## 1. Introduction

In 2006, EU member states (based on European Commission Regulation, 1831/2003) banned the use of antibiotics in animal fodder for fear of the risk of their trace presence in milk and meat [1]. In addition, consumers have recently paid special attention to the origin and quality of the animal products they consume. This results in increased demand for traditional and organic products. There is also a growing awareness of the dangers arising from the use of agricultural chemicals that can have negative impact on people’s health. The described phenomena caused a heightened interest in alternative growth stimulants of natural origin as well as the possibilities of using herbs in the prevention and treatment of animals.

Herbs can be a perfect supplement in the animal diet. In the literature on various animal species, it has been shown that the active substances they contain, such as: alkaloids, glycosides, phenols, saponins, tannins, coumarins and essential oils, have antibacterial, anti-inflammatory and antiparasitic properties [2,3,4,5]. In addition, they improve the functioning of the immune system, regulate the appetite and feed intake of animals, regulate the functioning of the digestive system, stimulate the metabolism of the body as well as quality parameters of meat [4]. These properties mean that herbs are increasingly being introduced into granulated fodders for fattening rabbits.

Daily weight gain is one of the most important indicators of the profitability commercial production of rabbits. There are many factors affecting this, and in particular the selection of the breed and ensuring optimal environmental conditions, among which the most important is proper nutrition. To increase the bioavailability of nutrients from food, various herbal supplements can be used as natural growth stimulants [6]. These additives often improve the palatability of food, thereby increasing its intake, and ultimately affecting the increase in animal production. According to literature data, herbs that have a beneficial effect on the daily growth of rabbits include, among others: oregano (*Origanum vulgare*) [7,8], sage (*Salvia officinalis* L.) [9] and thyme (*Thymus vulgaris* L.) [10].

Another a very important element in rearing rabbits is to protect their growing organism against the effects of various disease factors. Available research results confirm the bacteriostatic, antifungal and coccidiostatic effects of such herbs as: oregano (*Origanum vulgare*) [11], thyme (*Thymus vulgaris* L.) [12], fennel (*Foeniculum vulgare Mill*) [13] and black cumin (*Nigella sativa*) [14].

The quality of rabbit meat, its structure, chemical composition, as well as colour and taste are influenced by many factors, including gender, age, body weight before slaughter, as well as breed and, importantly, nutrition [15]. Properly selected feed phytoadditives have a positive effect on the smell and palatability of the feed, as well as improving the appetite of animals, which is conducive to increasing feed intake: this, in turn, may translate into production parameters [4]. Due to the structure of their digestive tract and the specificity of digestion, rabbits are very sensitive to changes in the feed used, hence the importance of carefully examining the impact of herbal feed additives on their health, welfare, and also on the quality of the final product, in this case rabbit meat.

Previous studies have confirmed that herbs and spices effectively conserve and improve the quality of meat and meat products by acting as antioxidants. Oxidative processes are one of the basic mechanisms of deterioration of the quality of meat and meat products, because they depreciate its taste, colour and nutritional value, and that shorten its shelf life [16]. Studies have shown that oil of oregano (*Origanum vulgare*) prevents lipid oxidation in rabbit meat [17]. It has also been demonstrated that the addition of milk thistle (*Silybum marianum (L.) Gaertner*) to food increases the acidity of *m. longissimum lumborum* of rabbits meat one day after slaughter [18].

In the literature regarding herbal supplements to rabbit fodder, there is not much information on the use of nettle or fenugreek in their nutrition. Both of these herbs are valuable sources of vitamins and minerals. They regulate the digestive system and stimulate appetite. They have a positive impact on the functioning of the immune system and they exhibit antibacterial activity [19,20]. Nettle improves biochemical, haematological, and immunological parameters [21]. Fenugreek stimulates the body to produce mucus, which allows the removal of allergens from the respiratory system and toxins from the urinary tract. Numerous studies have shown that fenugreek can also have anti-inflammatory, analgesic, and antipyretic properties [22,23].

The purpose of the study was to analyse the effect of nettle (*Urtica dioica* L.) leaves and fenugreek (*Trigonella foenum-graecum* L.) seeds as additives to fodder in order to improve post-slaughter traits and quality parameters Termond White of rabbit meat.

## 2. Materials and Methods

### 2.1. Animals

The experiment was conducted under standardized conditions at the Experimental Station of the Department of Genetics, Animal Breeding and Ethology, University of Agriculture in Krakow. The experimental material was Termond White rabbits (n = 60; 30♂, 30♀). Until weaning, rabbits with their mothers were housed in wooden cages placed in a heated hall equipped with a water trough installation (nipple drinkers), a lighting installation (14L:10D), and a forced ventilation system. From weaning (on the 35th day of life) until the 84th day of life, the animals were kept in wire metal cages intended for commercial rearing of rabbits (2 rabbits per cage). In the experiment, we used 20 does and from each litter, whereas 1 male and 1 female were randomly assigned to each experimental group. Three experimental groups were created: control group (n = 20; 10♂ and 10 ♀) was fed ad libitum with complete feed. The mixture for this group consisted of: corn, bran, wheat, dried alfalfa, dried alfalfa, soybean meal, sunflower meal, dicalcium phosphate, calcium carbonate and vitamin-mineral premix (Table 1). The nettle leaves (crude protein: 28%, crude fat: 3.3%, crude fiber: 26%) and fenugreek seeds (crude protein: 28.5%, crude fat: 6.5%, crude fiber: 10.2%) were bought by a feed manufacturer FHP Barbara Ltd. These additives were mashed, mixed with all ingredients and then pelleted.

The animals from the group feeding with nettle addition (n = 20; 10♂ and 10♀) were fed a complete mixture added with 1% of nettle (*Urtica dioica* L.) leaves. Rabbits from group feeding with fenugreek addition (n = 20; 10♂ and 10♀) were fed with a complete mixture added with 1% of fenugreek (*Trigonella foenum-graecum* L.) seeds. The weight at weaning (mean with standard deviation) were 1011 g ± 205 g in control group, 1125 g ± 274 g in group fed with nettle leaves addition and 1087 g ± 218 g in group fed with fenugreek seeds addition. Water was available ad libitum. Animals were slaughtered on the 84th day of life. Experiment was conducted under a permit from the Local Ethics Commission (agreement no 267/2018).

### 2.2. Carcass Traits

After 24 h fasting with a constant access to water, slaughter body weight was recorded, and animals were subsequently slaughtered. The animals were stunned, immediately bled, pelted and eviscerated. After slaughter, hot carcass weight was recorded, and after 24 h storage at 4 ${}^{\circ }$C, chilled carcass weight was also recorded. After this time, the carcasses were divided into three basic parts: fore part, middle part, and hind part. These parts were weighed. All measurements were made using electronic scales of Łucznik KS-205 (Galeria Łucznik Co. Ltd., Wrocław, Poland, e = 0.1).

Dressing out percentages were also calculated. The dressing out percentage hot (DPH) was defined as 100 × (HCW/SW), and dressing out percentage chilled as DPC = 100 × (CCW/SW), where HCW is hot carcass weight, SW is slaughter weight and CCW is chilled carcass weight. The processes of slaughter and dissection were conducted using the methods described by Blasco et al. (1993) [24].

### 2.3. Colour Measurement

The lightness [L${}^{*}$], redness [a${}^{*}$] and yellowness [b${}^{*}$] of meat were measured using chroma meters of Minolta CR-410 brand (Minolta Co. Ltd., Osaka, Japan). The colour was recorded 45 min after slaughter and 24 h after slaughter on the middle part (*m. longissimus lumborum*) and the hind part (*m. biceps femoris*).

### 2.4. Texture Analysis

Cylindrical samples from the *m. longissimus lumborum* were cut from the right half of the loin. The samples were vacuum packed in the foil used for food storage and then frozen for 72 h at −18 ${}^{\circ }$C, then thawed at room temperature and boiled in a water bath at 80 ${}^{\circ }$C for 40 min. Shear force and texture parameters were measured using TA.XTplus Texture Analyser (Stable Micro Systems Co. Ltd., Godalming, UK). Shear force was measured from three cylindrical samples (15 mm diameter, 15 mm height) using a Warner–Bratzler attachment and a triangular notch in the blade. Meat samples were cut perpendicular to the direction of muscle fibres. Blade speed during the test was 2 mm/s. Texture (hardness, springiness, cohesiveness, chewiness) was analysed using the attached cylinder, 50 mm in diameter. The three samples were subjected to a double pressing test, applying a force of 10 g to 70 % of their height. Cylinder speed was 5 mm/s, and the interval between presses was 5 s.

### 2.5. Microstructural Analysis

Within 24 h post mortem, muscle samples for microstructural analysis were taken from the right side of the carcass from the *m. longissimus lumborum* at the level of the 1st lumbar vertebra from the middle part of the muscle. Muscle samples were cut into 1 cm${}^{3}$ pieces (parallel to the muscle fibres) and frozen in isopentane that was cooled using liquid nitrogen and stored at −80 ${}^{\circ }$C until subsequent analyses. Samples were mounted on a cryostat chuck with a few drops of tissue-freezingmedium (Tissue-Tek; Sakura Finetek Europe, Zoeterwoude, The Netherlands). Transverse Sections (10-m thick) were cut at −20 ${}^{\circ }$C in a cryostat (Slee MEV, Mainz, Germany). To distinguish muscle fibre types (I, IIA and IIB), a modified combined method of NADH-tetrazolium reductase activity was used and immunohistochemical determination of the slow myosin heavy chain on the same section with monoclonal antibodies against the skeletal slow myosin heavy chain was performed for 1 h at RT (NCL-MHCs, clone WB-MHCs Leica Biosystems, Germany, dilution 1:80) [25]. The reaction was visualized using the NovoLinkTM Polymer Detection System (Leica, Nussloch, Germany) Finally, all sections were dehydrated in a graded series of ethyl alcohol, cleared in xylene and mounted in DPX mounting medium (Fluka, Buchs, Switzerland). A minimum of 300 fibres had been counted in each section using a NIKON E600 light microscope. The percentage and diameter of muscle fibre types were quantified with an image analysis system using the Multi Scan v. 14.02 (Computer Scanning Systems Ltd., Warsaw, Poland) computer software.

### 2.6. Statistical Analysis

Analysis of variance using the MIXED procedure of SAS [26] was performed in order to evaluate the effect of feeding and gender. The following linear model was used:(1)$$Y_{ijk}=\mu +FE_{i}+SEX_{j}+{\left(FE\times SEX\right)}_{ij}+\epsilon_{ijk}$$
where:$Y_{ijk}$—post-slaughter traits or meat quality parameters,μ—overall mean,$FE_{i}$—effect of *i*-th feeding (*i* = 1, 2, 3),$SEX_{j}$—effect of *j*-th gender (*j* = 1, 2),${\left(FE\times SEX\right)}_{ij}$—effect of interaction between feeding and gender,$\epsilon_{ijk}$—residual effect.

The significance of differences was determined using the Tukey-Kramer test. All *p*-values less than 0.05 were considered as statistically significant.

## 3. Results

In most cases, the effect of interaction between feeding and gender was not significant (*p* > 0.05). The significant interaction (*p* < 0.05) was only observed for liver weight, dressing out percentage hot, and dressing out percentage chilled.

### 3.1. Post-Slaughter Traits

Table 2 shows the effect of feeding and gender on post-slaughter traits. Rabbits fed with pellets with the addition of nettle were characterized by a higher slaughter weight and a weight of hot and cold carcasses compared to the rabbits from the control group and the group fed with pellets with fenugreek. In addition, it was observed that the animals fed pellets with nettle had a higher weight of the lungs, kidney and head as well as a higher weight of the fore, middle and hind part of carcasses compared to other two groups. The animals from the control group had a significantly lower heart weight compared to the animals fed with nettle feed. The carcasses of animals fed with fenugreek and nettle had a higher percentage of the fore and hind parts compared to the carcasses of the animals from the control group. There were no significant differences in the percentage of the middle part between the food groups studied (Table 2). The female carcasses were characterized by a significantly higher percentage of the middle part compared to the male carcasses. There were no significant differences between the genders for other post-mortem traits (*p* > 0.05; Table 2).

### 3.2. Meat Quality Traits

The second studied group of traits was meat quality traits. The least squares means (LSM) of colour measurement traits are presented in Table 3. For most colour measurement traits, the differences depending on the feeding regimes had been significant (*p* < 0.05), except the lightness (L${}_{24}^{*}$) and yellowness (b${}_{24}^{*}$) 24 h after slaughter on the *m. biceps femoris*, and lightness (L${}_{24}^{*}$) on the *m. longissimus lumborum*. The carcasses of the rabbits receiving pellets with fenugreek were characterized by a significantly higher L${}^{*}$ value measured for 45 min after slaughter on the *m. biceps femoris* in comparison with the other groups. However, 45 min and 24 h after slaughter, the meat of rabbits receiving fodder with nettle leaves was characterized by a higher value of redness compared to the other groups (a${}_{45}^{*}$ and a${}_{24}^{*}$), i.e., control group and fenugreek group. The higher value of yellowness (b${}_{45}^{*}$) on the *m. biceps femoris* was observed in the control group in comparison with the other groups (Table 3). The carcasses of the rabbits fed with pellets with the addition of nettle had a significantly lower L${}^{*}$ value measured 45 min after slaughter on the *m. longissimus lumborum* compared to the other groups. Moreover, it was found that the meat of animals from this group was characterized by the highest value of the red component (a${}_{45}^{*}$), 45 min after slaughter. The lowest value of redness (a${}_{45}^{*}$) of *m. longissimus lumborum* 45 min after slaughter was found in the control group. In addition, 45 min after slaughter, animals coming from control group had the lower value of yellowness (b${}_{45}^{*}$). The higher value of redness (a${}_{24}^{*}$) and yellowness (b${}_{24}^{*}$) on the *m. biceps femoris* was observed in the group fed with pellets with the addition of nettle in comparison with the other groups (Table 3). Additionally, in this group of traits, the differences between sexes were non significant (Table 3; *p* > 0.05).

The LSM of texture and muscle fibre traits are presented in Table 4. The effect of feeding and gender on texture traits such as shear force, hardness, springiness, cohesiveness and chewiness were non significant (Table 4; *p* > 0.05).

The last analysed group of traits were muscle fibre parameters of *m. longissimus lumborum* (Table 4). The results obtained in this study showed that feeding had no effect (*p* > 0.05) on muscle fibre diameter, but that it affected the muscle fibre type I percentage (*p* < 0.05). Thus, the group fed with pellet containing nettle leaves had higher percentage of type I muscle fibres than the control group. The effect of gender on muscle fibre traits was non significant (Table 4; *p* > 0.05).

## 4. Discussion

The literature on the nutrition of rabbits regarding the addition of fenugreek seeds and nettle leaves to their feed and the impact there of on their growth, on post-slaughter and meat quality traits is very sparse. Tag El-Din et al. (2016) who studied the impact of the addition of fenugreek, marjoram or both concluded that it had a big impact on the growth and post-slaughter traits of New Zealand White rabbits. The authors confirmed that the addition of these herbs to the feed improved the ultimate body weight of the animals (at week 14), but they did not prove that the herbs improved post-slaughter traits such as carcass weight, heart and liver weight and dressing out percentage [27]. Dawood et al. (2015) also failed to confirm that the 1.5% or 3.0% addition of fenugreek seeds improved body weight in male rabbits [28]. Zeweil et al. (2015) indicated that diets containing 0.6% fenugreek seeds had no significant effect on carcass weight or organ relative weight compared to the control group [29]. Studies on broiler chickens showed that the addition of fenugreek seeds (0.5% or 1.5%) to the feed had a significant effect on body weight at 6 weeks, body weight gain, feed conversion ratio, protein efficiency ratio, feed consumption and efficiency of energy utilization in broiler chicks [30]. El-Gharmy et al. (2004) found that the addition of fenugreek to chicken fodder (1.5%) lead to a significantly heavier live body weight and body weight gain than those fed on control diet [31]. Weerasingha and Atappatu (2013) demonstrated that 1 % dietary fenugreek seed powder had a positive growth effect in chicken broilers but more than 1% of fenugreek seed powder in the chicken fodder had a negative impact on feed consumption and growth [32]. However, Keshavarz et al. (2014) indicated that the supplementation of starter and finisher diets with the addition of essential oil from nettle leaves and nettle leaves powder had no significant effect on the growth parameters. Effects of dietary treatments on carcass traits were also non significant. In contrast, the results of aforementioned authors showed that addition of essential oil from nettle leaves or nettle leaves powder had significant positive effect on liver weight, gizzard weight, and lungs weight. The other internal organs were not impacted by diets supplemented with nettle leaves powder or nettle leaves essential oil [33]. Safamehr et al. (2012) found that 1–2% nettle leaves supplementation had a positive effect on growth and carcass traits of broiler chickens [34]. Szewczyk et al. (2006) showed that the addition of nettle leaves’ extract to pigs’ feed ( 500 mg/kg of feed or 1000 mg/kg of feed) did not significantly affect their body weight, weight gain, meat content in primal cuts, whole carcass or backfat thickness. These authors found that supplementing the feed with 1000 mg/kg with dried water extract of nettle leaves would significantly improve the animals cold dressing percentage [35].

The most important parameter in the production of rabbits is dressing out percentage. This parameter should be over 50%, which is one of the conditions for profitability of breeding. Correct dressing out percentage is important for slaughterhouses as well as for consumers. The commercialization of well-muscled rabbit carcasses improves the demand for this product. The factors influencing the rabbit carcasses’ quality include the selection of the appropriate breed and proper feeding. In this study the DPH had proper values regardless of diet and gender. It has been proven in many studies that certain herbs and spices improve the growth and quality of rabbit meat [7,17,18,36].

The colour of meat is one of the basic distinguishing features of its technological and culinary quality and is one of the first and most important distinguishing features of its consumer evaluation. In the presented work, the colour of the meat was assessed on the CIE L${}^{*}$ a${}^{*}$ b${}^{*}$ scale, where these distinguishing features determined the lightness of colour, the share of redness, and the share of yellowness, respectively. Research by Bízková and Tůmová (2010) and Maj et al. (2012) indicate that the colour of rabbit’ muscles can be influenced by a number of factors, such as the age of the animal, the experienced stress, and the genotype of the individual [15,37]. Nutrition can also play an important role. The myoglobin is also of great importance, as it may be present in fresh meat of several forms: oxyhemoglobin, deoxymyoglobin and metmyoglobin. The presence of these pigments in the muscle is correlated with the amount of oxygen present and with the oxygen changes in the original pigment, which also contribute to the specific colour of the meat [38]. Similarly, Feldhusen et al. (1995) determined that the colour of meat depends both on the amount and the degree of oxidation of heme pigments [39]. In the present research, the plant supplement in the form of fenugreek did not have a significant effect on all the analysed colour parameters. On the other hand, enrichment of the feed with the addition of nettle significantly increased the value of the a${}^{*}$ and b${}^{*}$ colour parameters compared to the control group and the group fed with fenugreek. At the moment, however, there are no similar studies in the professional literature whose results could have been compared with those obtained in this paper. However, there are studies that have been conducted on other species of animals, or those fed with other herbal supplements, where the effects of nutrition on the colour of meat have been analysed. And so, the effect of the feed additive in the form of cranberry extract on the colour of rabbit meat was indirectly confirmed in the studies by Kone et al. (2003) [40]. Cranberry significantly increased the a${}^{*}$ colour parameter, but on the other hand it did not affect the b${}^{*}$ colour parameter. In turn, Batkowska and Brodacki (2011), having analysed the impact of the feeding system and the maintenance of turkeys on the colour of meat, showed that turkeys fed with feed additives such as nettle, clover and alfalfa were characterized by a higher b${}^{*}$ colour index [41]. Dabbou et al. (2017), on the other hand, have analysed the effect of the feed supplement in the form of blueberry on the quality traits of the rabbit’s hind leg muscle, and found that this plant did not have a significant effect on the values of colour parameters [42]. Rudy (2010) showed that rabbit meat is classified as white meat, and its bright colour is a natural phenomenon, resulting, among other things, from the reduced content of myoglobin in the muscles of these animals [43]. The colour of meat largely depends on the composition of the muscle fibres [44,45]. Hence, the light colour of rabbit muscles is related mainly to the high proportion of type IIB fibres in the muscle fibre composition, which are characterized by a low content of myoglobin and a high content of myofibrils [46]. These assumptions are also confirmed by the present research, which showed that regardless of the analysed nutritional group, rabbit muscles are characterized by a very high percentage of type IIB muscle fibres. On the other hand, higher values of colour parameter a${}^{*}$ in the meat of rabbits fed with nettle may be related to a significantly higher share of type I muscle fibres in these animals compared to the control group. These suggestions are confirmed by earlier studies by Ryu and Kim (2005), in which significant positive correlations were found between the percentage of type I muscle fibres and the values of the colour parameter a${}^{*}$ [44]. We need to remember that type I muscle fibres are fibres characterized by a high content of myoglobin [46], which is responsible, among others, for the red colour of these fibres, and this translates into the colour of the entire muscle.

The microstructural analysis carried out in this study showed no effect of nutrition, either with fodder enriched with fenugreek or with nettle, on the diameter of all analysed types of muscle fibres. The size of the muscle fibres depends mainly on the type of muscle, and this may explain the lack of differences in the size of the muscle fibres, shown in this study, between the analysed groups of animals. Moreover, due to the fact that this is the first study to analyse the effect of herbal additives such as fenugreek or nettle on the microstructure of rabbit muscles, it is difficult to precisely explain the obtained results. It can only be noted that the size of the muscle fibres analysed in this study, *m. longissimus lumborum* in rabbits of all analysed food groups, is similar to the results obtained in previous studies that analysed the microstructure of the muscles of rabbits of various genotypes [47].

In the present study, analysing the microstructure of *m. longissimus lumborum* in rabbits, no significant effect of feeding with fodder enriched with fenugreek on the percentage of muscle fibres of type I, IIA and IIB was demonstrated, which may indicate that the feed additive used does not have a negative effect on the composition of muscle fibres. On the other hand, different results were obtained in the case of common nettle, where a significantly higher percentage of type I muscle fibres, the most favourable in terms of meat quality parameters, were recorded in the muscles of rabbits fed with 2% of this herb compared to the muscles of control rabbits. Type I muscle fibres are fibres with oxidative metabolism. Therefore, the increased percentage of this type of fibres in the composition of muscle fibres proves the more oxidative nature of the muscles of rabbits fed with a mixture including nettle compared to control rabbits, and this may have a significant impact on such parameters of meat quality as: pH, colour, water holding capacity or tenderness [44,45]. Kwasiborski et al. (2008) have demonstrated showed that muscles characterized by oxidative metabolism are characterized by higher pH values, in contrast to muscles with glycolytic fibres, which have a lower pH [48]. In turn, Larzul et al. (1997) and Ryu and Kim (2005) found that muscles with a higher percentage of type I fibres exhibited less free leakage compared to muscles with a higher proportion of type IIB fibres [44,49]. The composition of muscle fibres is also translated into the color of the meat [44,50].

The lack of differences in the shear force values between the groups of rabbits analysed in this study may result from the fact that tenderness is also determined by other factors, not only the composition of muscle fibres. Tenderness is a feature influenced by, among others, the species, gender of the animal, vital factors, handling of the carcass after slaughter, as well as water absorption and the amount of water contained in the meat, as well as the acidity of the meat [51], which has a decisive influence on the activity of the calpain system involved in post-slaughter proteolysis [52].

## 5. Conclusions

Summing up, it can be stated that feeding rabbits with herbal supplements, such as nettle (*Urtica dioica* L.) or fenugreek (*Trigonella foenum-graecum* L.), has no negative impact on technological parameters of the meat or the microstructure of the muscles. On the contrary, in the case of nettle, one can speak of a beneficial effect of this herbal supplement on the post-slaughter traits and microstructure of the muscles, as evidenced by the greater share of type I muscle fibres in the composition of muscles, e.g., *longissimus lumborum*, indicating a more oxidative nature of these muscles, which in turn may translate into the quality of rabbit meat.

## Figures and Tables

**Table 1 animals-11-01566-t001:** Ingredients and chemical composition of the control and experimental feed (according to feed manufacturer FHP Barbara Ltd.).

	Feeding		
Components	Control Group ${}^{a}$	Nettle Leaves ${}^{b}$	Fenugreek Seeds ${}^{c}$
	Ingredients (%)		
Wheat	29.58	28.58	28.58
Corn	24.50	24.50	24.50
Bran	15.00	15.00	15.00
Sunflower meal	11.00	11.00	11.00
Dried alfalfa	10.00	10.00	10.00
Soybean meal	7.00	7.00	7.00
Vitamin-mineral premix	1.50	1.50	1.50
Calcium carbonate	0.80	0.80	0.80
Dicalcium phosphate	0.62	0.62	0.62
Nettle leaves	0.00	1.00	0.00
Fenugreek seeds	0.00	0.00	1.00
	Chemical composition (%)		
Crude protein	16.40	16.55	16.50
Metabolic energy [MJ/kg]	10.11	10.16	10.19
Crude fibre	9.22	9.12	9.15
Crude ash	4.84	4.93	4.87
Crude fat	2.70	2.75	2.80
Calcium	0.77	0.77	0.77
Lysine	0.66	0.66	0.68
Phosphorus	0.63	0.63	0.63
Methionine	0.29	0.29	0.30
Sodium	0.24	0.24	0.24

^a^ Pelleted commercial diet; ^b^ Diet with nettle leaves (*Urtica dioica* L.); ^c^ Diet with fenugreek (*Trigonella foenum-graecum* L.) seeds.

**Table 2 animals-11-01566-t002:** Least square means (LSM) with standard errors (SE) of post-slaughter traits, by feeding or gender.

	Feeding									Gender				
	Control Group			Nettle Leaves ${}^{1}$			Fenugreek Seeds ${}^{2}$			Male			Female	
Trait	LSM	SE		LSM	SE		LSM	SE		LSM	SE		LSM	SE
Slaughter weight [g]	2380.21${}^{A}$	58.24		2329.75${}^{A}$	58.24		2927.75${}^{B}$	57.94		2541.20	47.47		2550.60	47.47
Hot carcass weight [g]	1211.60${}^{A}$	35.35		1191.57${}^{A}$	35.35		1487.45${}^{B}$	35.17		1317.12	28.81		1276.62	28.81
Liver [g]	68.32${}^{A}$	3.70		62.75${}^{A}$	3.70		92.50${}^{B}$	3.68		71.09	3.01		77.96	3.01
Lungs [g]	18.19${}^{A}$	1.16		21.52${}^{A}$	1.16		27.20${}^{B}$	1.15		22.29	0.94		22.31	0.94
Heart [g]	7.37${}^{A}$	0.45		8.89${}^{AB}$	0.45		9.60${}^{B}$	0.45		8.84	0.37		8.40	0.37
Kidneys [g]	15.34${}^{A}$	0.62		14.27${}^{A}$	0.62		18.20${}^{B}$	0.62		16.49	0.51		15.38	0.51
Head [g]	120.77${}^{A}$	2.39		122.55${}^{A}$	2.39		135.80${}^{B}$	2.38		128.86	1.95		123.88	1.95
Fore part weight [g]	469.70${}^{A}$	14.83		481.75${}^{A}$	14.83		610.70${}^{B}$	14.75		531.90	12.09		509.54	12.09
Middle part weight [g]	241.78${}^{A}$	9.43		237.10${}^{A}$	9.43		310.25${}^{B}$	9.38		260.48	7.69		265.60	7.69
Hind part weight [g]	440.53${}^{A}$	11.38		436.68${}^{A}$	11.38		529.70${}^{B}$	11.33		476.16	9.28		461.77	9.28
Chilled carcass weight [g]	1152.00${}^{A}$	33.86		1155.53${}^{A}$	33.86		1450.65${}^{B}$	33.69		1268.54	27.60		1236.91	27.60
DPH ${}^{3}$ [%]	50.84	0.50		51.16	0.50		50.72	0.50		51.76${}^{A}$	0.41		50.06${}^{B}$	0.41
DPC ${}^{4}$ [%]	48.42	0.54		49.60	0.54		49.47	0.54		49.84${}^{A}$	0.44		48.48${}^{B}$	0.44
Fore part of the carcass [%]	40.71${}^{A}$	0.29		41.70${}^{B}$	0.29		42.09${}^{B}$	0.28		41.82	0.23		41.18	0.23
Middle part of the carcass [%]	20.99	0.31		20.45	0.31		21.33	0.31		20.48${}^{A}$	0.25		21.37${}^{B}$	0.25
Hind part of the carcass [%]	38.31${}^{A}$	0.26		37.86${}^{B}$	0.26		36.58${}^{B}$	0.26		37.70	0.21		37.46	0.21

${}^{A,B}$ values in the same row within feeding group or sex with different superscripts differ significantly (*p* < 0.05). ${}^{1}$ Diet with nettle leaves (*Urtica dioica* L.). ${}^{2}$ Diet with fenugreek (*Trigonella foenum-graecum* L.) seeds. ${}^{3}$ dressing out percentage hot. ${}^{4}$ dressing out percentage chilled.

**Table 3 animals-11-01566-t003:** Least square means (LSM) with standard errors (SE) of lightness (L${}^{*}$), redness (a${}^{*}$) and yellowness (b${}^{*}$) of meat, by feeding and gender.

	Feeding									Gender				
	Control Group			Nettle Leaves ${}^{1}$			Fenugreek Seeds ${}^{2}$			Male			Female	
Trait	LSM	SE		LSM	SE		LSM	SE		LSM	SE		LSM	SE
*m. biceps femoris*														
L${}_{45}^{*}$	50.67${}^{A}$	0.59		53.87${}^{B}$	0.59		49.34${}^{A}$	0.59		51.15	0.48		51.44	0.48
a${}_{45}^{*}$	2.61${}^{A}$	0.23		2.69${}^{A}$	0.23		3.63${}^{B}$	0.23		2.96	0.19		2.98	0.19
b${}_{45}^{*}$	2.50${}^{A}$	0.32		0.36${}^{B}$	0.32		1.39${}^{B}$	0.31		1.31	0.26		1.52	0.26
L${}_{24}^{*}$	55.91	0.45		56.54	0.45		55.57	0.45		55.74	0.37		56.27	0.37
a${}_{24}^{*}$	3.17${}^{A}$	0.25		3.41${}^{A}$	0.25		4.81${}^{B}$	0.25		4.02	0.21		3.57	0.21
b${}_{24}^{*}$	3.90	0.28		3.08	0.28		3.56	0.28		3.20	0.23		3.83	0.23
*m. longissimus lumborum*														
L${}_{45}^{*}$	60.56${}^{A}$	0.70		59.59${}^{A}$	0.70		57.00${}^{B}$	0.69		59.19	0.57		58.91	0.57
a${}_{45}^{*}$	0.25${}^{A}$	0.63		3.64${}^{B}$	0.63		6.19${}^{C}$	0.63		2.95	0.52		3.78	0.52
b${}_{45}^{*}$	−4.71${}^{A}$	0.70		−1.00${}^{B}$	0.70		0.20${}^{B}$	0.70		−2.58	0.57		-1.09	0.57
L${}_{24}^{*}$	55.97	0.63		57.37	0.63		56.53	0.63		56.65	0.51		56.60	0.51
a${}_{24}^{*}$	4.88${}^{A}$	0.47		5.32${}^{A}$	0.47		8.23${}^{B}$	0.47		6.13	0.38		6.15	0.38
b${}_{24}^{*}$	1.98${}^{A}$	0.46		1.42${}^{A}$	0.46		4.23${}^{B}$	0.46		2.40	0.37		2.69	0.37

${}^{A,B,C}$ values in the same row within feeding group or sex with different superscripts differ significantly (*p* < 0.05). ${}^{1}$ Diet with nettle leaves (*Urtica dioica* L.) ${}^{2}$ Diet with fenugreek (*Trigonella foenum-graecum* L.) seeds.

**Table 4 animals-11-01566-t004:** Least square means (LSM) with standard errors (SE) of texture and muscle fibre traits (—diameter and %—percentage), by feeding and gender.

	Feeding									Gender				
	Control Group			Nettle Leaves ${}^{1}$			Fenugreek Seeds ${}^{2}$			Male			Female	
Trait	LSM	SE		LSM	SE		LSM	SE		LSM	SE		LSM	SE
Shear force [kg]	1.61	0.07		1.58	0.07		1.76	0.07		1.65	0.06		1.65	0.06
Hardness [kg]	10.84	0.40		9.93	0.40		9.68	0.40		10.16	0.32		10.14	0.32
Springiness	0.48	0.12		0.65	0.12		0.47	0.12		0.59	0.10		0.47	0.10
Cohesiveness	0.42	0.01		0.41	0.01		0.42	0.01		0.41	0.01		0.42	0.01
Chewiness [kg]	2.22	0.11		1.86	0.11		1.90	0.11		1.96	0.09		2.03	0.09
øI	35.76	1.46		32.79	1.46		34.04	1.64		35.19	1.27		33.21	1.22
øIIA	34.04	1.14		31.22	1.15		31.68	1.28		33.09	0.99		31.53	0.95
øIIB	46.63	1.22		42.89	1.21		45.20	1.37		45.73	1.05		44.08	1.02
%I	3.03${}^{A}$	0.32		3.25${}^{AB}$	0.33		4.72${}^{B}$	0.35		3.72	0.27		3.63	0.26
%IIA	7.57	0.61		7.11	0.61		7.65	0.69		7.69	0.53		7.39	0.51
%IIB	89.40	0.81		89.64	0.80		87.63	0.90		88.80	0.70		88.98	0.67

${}^{A,B}$ values in the same row within feeding group or sex with different superscripts differ significantly (*p* < 0.05). ${}^{1}$ Diet with nettle leaves (*Urtica dioica* L.) ${}^{2}$ Diet with fenugreek (*Trigonella foenum-graecum* L.) seeds.

## Data Availability

All data are available from the authors’ database.

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
