# Peer review of "Effect of a Diet Supplemented with Nettle (Urtica dioica L.) or Fenugreek (Trigonella foenum-graecum L.) on the Post-Slaughter Traits and Meat Quality Parameters of Termond White Rabbits"

_animals, 2021, doi:10.3390/ani11061566_

Round 1
Reviewer 1 Report
The manuscript is very interesting from the scientific perspective, and the findings may have important practical implications. The study investigated the effect of diet supplemented with nettle or fenugreek on post-slaughter traits and meat quality parameters of rabbits. Below are my specific comments that may help improve the manuscript.
- I believe the title of the manuscript should be modified. The addition "preliminary research" is unnecessary. However, I leave the final decision to the authors.
- Please consider cite the works "In polish". In my opinion, there are too many of them at this work.
- Information from lines 103-107 is redundant. They are given in table 1.
- 1. Convert (% per kg) to (%). Please provide the chemical content of nettle leaves and fenugreek seeds.
- Move lines 263-271 to the Introduction chapter
- Conclusion, line 352: please change the order. First, the nettle, as in tab. 1.
In general, the study has high scientific merit. The manuscript can be accepted for publication after revision, addressing the above comments.
Author Response
Thank you for reviewing our manuscript and we greatly appreciate the Reviewer’s suggestions and many valuable comments which helped us to correct the manuscript. The revision was made using “Track Changes” function (trackchanges packages) in LaTeX. However, due to problems with the application of the package, e.g. in the title of the paper or when SI units were used we added the comments in the text.
Comments:
I believe the title of the manuscript should be modified. The addition "preliminary research" is unnecessary. However, I leave the final decision to the authors.
Response:
The title of the manuscript was changed.
Comments:
Please consider cite the works "In polish". In my opinion, there are too many of them at this work.
Response:
Some of the cited papers have been changed or removed. However, three unforgettable papers written in Polish were left.
Comments:
Information from lines 103-107 is redundant. They are given in table 1.
Response:
The information was removed (see line 112).
Comments:
Convert (% per kg) to (%).
Response:
It was changed, i.e. (% per kg) to (%).
Comments:
Please provide the chemical content of nettle leaves and fenugreek seeds.
Response:
We have added information on the chemical composition of nettle leaves and fenugreek seeds (see lines 114-116). Detailed information about the chemical composition of these herbs will be included in our other publication (in the review), so we would not like to include them in this publication due to the risk of being accused of autoplagiarism.
Comments:
Move lines 263-271 to the Introduction chapter
Response:
Text was moved to Introduction chapter – see lines 74-82.
Comments:
Conclusion, line 352: please change the order. First, the nettle, as in tab. 1.
Response:
It was changed (see lines 355-356).
Reviewer 2 Report
In general the article is very interesting and well presented (introduction, results, conclusion). The article fill the scope of the journal. Anyway, there are some comments regarding the Material and methods and Discussion section:
Lines 94-113: no information are given regard an average weight of animals at weaning time. It is supposed that the average weight of different groups was homogenous. Please, add the average weight of animals at weaning. This information could help Authors to better explain the higher weight reached by the feengreek seeds group.
Please, indicate what is the origin of the data in Table 1. Are they from manufacture company? or are they from laboratory analyses? In the second case, please indicate laboratory procedures used.
Last missing information is regarding the supplements used. Are they both commercial products? Or collected and prepared (maybe dried) by Authors? In the second case, please indicate laboratory procedures used.
Authors used (in lines 109 and 110) the term "enriched". Please, specify what the term means. Were they mixed and mashed with all ingredients and then pelleted?
Author Response
Thank you for reviewing our manuscript and we greatly appreciate the Reviewer’s suggestions and many valuable comments which helped us to correct the manuscript. The revision was made using “Track Changes” function (trackchanges packages) in LaTeX. However, due to problems with the application of the package, e.g. in the title of the paper or when SI units were used we added the comments in the text.
Comments:
Lines 94-113: no information are given regard an average weight of animals at weaning time. It is supposed that the average weight of different groups was homogenous. Please, add the average weight of animals at weaning. This information could help Authors to better explain the higher weight reached by the feengreek seeds group.
Response:
We added information about average weight of animals at weaning time (see lines 121-123).
Comments:
Please, indicate what is the origin of the data in Table 1. Are they from manufacture company? or are they from laboratory analyses? In the second case, please indicate laboratory procedures used.
Response:
We added this information in the title of Table 1.
Comments:
Last missing information is regarding the supplements used. Are they both commercial products? Or collected and prepared (maybe dried) by Authors? In the second case, please indicate laboratory procedures used.
Response:
We added this information (see lines 114-116).
Comments:
Authors used (in lines 109 and 110) the term "enriched". Please, specify what the term means. Were they mixed and mashed with all ingredients and then pelleted?
Response:
Changed the word "enriched" to "added" (see lines 118 and 120). We added this information (see lines 116-117).
This manuscript is a resubmission of an earlier submission. The following is a list of the peer review reports and author responses from that submission.